# CONTINUAL LEARNING VIA PRINCIPAL COMPONENTS PROJECTION

## ABSTRACT

Continual learning in neural networks (NN) often suffers from *catastrophic forgetting*. That is, when learning a sequence of tasks on an NN, the learning of a new task will cause weight changes that may destroy the learned knowledge embedded in the weights for previous tasks. Without solving this problem, it is difficult to use an NN to perform continual or lifelong learning. Although researchers have attempted to solve the problem in many ways, it remains to be challenging. In this paper, we propose a new approach, called *principal components projection* (PCP). The idea is that in learning a new task, if we can ensure that the gradient updates will only occur in the orthogonal directions to the input vectors of the previous tasks, then the weight updates for learning the new task will not affect the previous tasks. We propose to compute the principal components of the input vectors and use them to transform the input and to project the gradient updates for learning each new task. PCP does not need to store any sampled data from previous tasks or to generate pseudo data of previous tasks and use them to help learn a new task. Empirical evaluation shows that the proposed method PCP markedly outperforms the state-of-the-art baseline methods.

## 1 INTRODUCTION

Human brains have this extraordinary ability to learn a large number of tasks incrementally with high accuracy. Both the learning process of and the learned knowledge for the tasks have little negative interference of each other. In fact, the learned knowledge earlier can even help the learning of new tasks later. *Continual learning* or *lifelong learning* attempts to make the computer to do the same using neural networks (NNs). However, it has been proven challenging. One of the key problems or obstacles is the issue of *catastrophic forgetting* (CF). This problem was first reported by McCloskey & Cohen (1989). In their work on NN, they found that when learning a sequence of tasks incrementally in an NN, the training of each new task can cause the network to forget the information or knowledge learned from previous tasks. This usually means that the new task training may significantly modify the weights that have been learned for previous tasks, and thus degrade the model performance for the previous tasks. Without solving this problem, an NN is unsuitable for continual learning as it *forgets* the existing knowledge when it learns new things.

In the past few years, CF has attracted a great deal of research attention (Parisi et al., 2018a). A literature review will be given in Section 2. There are two main setups in continual learning: *class continual learning* (CCL) and *task continual learning* (TCL) (Dhar et al., 2019). The general term *continual learning* has been used for both, which has caused some confusion when comparing results in some cases. In the CCL setup, each task consists of one or more classes to be learned together and the learning process builds a single model for all tasks for classifying all the classes of the tasks learned so far. In testing, a test instance from any class may be presented to the model for it to classify. There is no prior information about which task the test instance comes from. Formally, CCL can be defined as follows.

**Class continual learning** (CCL). Given a sequence of datasets $\mathcal{D} = \{\mathcal{D}_1, \mathcal{D}_2, ..., \mathcal{D}_T\}$ of $T$ tasks. The dataset of task $k$ is $\mathcal{D}_k = \{(\boldsymbol{x}_k^i, y_k^i)_{i=1}^{n_k}\}$, where each $\boldsymbol{x}_k^i \in \boldsymbol{X}$ is a feature vector from task $k$ and $y_k^i \in \boldsymbol{Y}_k$ is the corresponding target class label of $\boldsymbol{x}_k^i$. All $\boldsymbol{Y}_k$'s are disjoint and $\bigcup_{k=1}^{T} \boldsymbol{Y}_k = \boldsymbol{Y}$. The goal of CCL is to construct a single prediction function or model $f : \boldsymbol{X} \to \boldsymbol{Y}$ that can correctly identify the target class $y$ for a given test instance $\boldsymbol{x}$.

In the TCL setup, each task is a separate classification problem (e.g., one task could be to classify different breeds of dogs and another task could be to classify different types of birds). Here, one model is built for each task, although all the models reside in the same neural network. In testing, the system knows which task each test instance belongs to and uses only the specific model for the task (dog or bird classification) to classify the test instance. Formally, TCL is defined as follows.

**Task continual learning** (TCL). Given a sequence of datasets $\mathcal{D} = \{\mathcal{D}_1, \mathcal{D}_2, ..., \mathcal{D}_T\}$ of $T$ tasks. The dataset of task $k$ is $\mathcal{D}_k = \{((\boldsymbol{x}_k^i, t^k), y_k^i)_{i=1}^{n_k}\}$, where each $\boldsymbol{x}_k^i \in \boldsymbol{X}$ is a feature vector from task $k$, $t^k \in \boldsymbol{T}$ is the indicator of task $k$, and $y_k^i \in \boldsymbol{Y}_k \subset \boldsymbol{Y}$ is the corresponding target class label. The goal of TCL is to construct a predictor $f : \boldsymbol{X} \times \boldsymbol{T} \to \boldsymbol{Y}$ to identify the correct class label $y \in \boldsymbol{Y}_k$ for $(\boldsymbol{x}, t^k)$ (the given test instance $\boldsymbol{x}$ from task $k$).

This paper works in the TCL setting and proposes a more effective technique to deal with CF. The proposed method is called *principal components projection* (PCP), which is inspired by the OWM technique in (Zeng et al., 2019). The idea is that in learning a new task, if we can ensure that gradient updates will occur only in the orthogonal directions to the input vectors of the previous tasks, then the weight updates for the new task will have little effect on the weights for the previous tasks.

However, OWM (Zeng et al., 2019) has a key weakness. Its weight update in learning the new task occurs only in the solution space of the previous tasks, which restricts the solution space for the new task. When the new task is quite different from previous tasks, the learning accuracy will suffer (see Section 3.1). The proposed PCP method deals with the problem. The key idea of PCP is that it computes the principal components of the input vectors (as in PCA) for each task at each level of the network and uses them to transform the input and then to project the gradient updates to the orthogonal directions in learning the new task so that the updates will not affect the previous tasks and can also go beyond the solution space of previous tasks (see the details in Section 3.2). This results in significantly better results.

Moreover, unlike many existing TCL methods (see Section 2), PCP does not need to store any sampled data from previous tasks or generate pseudo past data and use them to help learn a new task. Empirical evaluation shows that PCP outperforms existing stat-of-the-art baselines considerably.

## 2 RELATED WORK

Catastrophic forgetting (CF) has been dealt with since early 1990s. Li & Hoiem (2016) gave a good summary of the early work. The paper also proposed a new technique called LwF to deal with forgetting. It learns the parameters for the new task $\theta_n$ with the help of shared parameters for all tasks $\theta_s$ and parameters for old tasks $\theta_o$ without degrading much of the performance of the old tasks. The idea is to optimize $\theta_s$ and $\theta_n$ on the new task with the constraint that the predictions on the new task's examples using $\theta_s$ and $\theta_o$ do not shift much. Rusu et al. (2016) proposed a progressive neural network that retains a pool of pre-trained models and learns lateral connections among them. Kirkpatrick et al. (2017) proposed a model called EWC that quantifies the importance of weights to previous tasks, and selectively adjusts the plasticity of weights. This method is also used in (Schwarz et al., 2018). Motivated by EWC (Kirkpatrick et al., 2017), Zenke et al. (2017) measured the synapse consolidation strength in an online fashion and used it as regularization to deal with CF. Rebuffi et al. (2017) retained an exemplar set that best approximates the previous tasks to help train the new task. Similar ideas were also used in (Lopez-Paz & Ranzato, 2017; Chaudhry et al., 2019; Ans et al., 2004). (Shin et al., 2017; Kamra et al., 2017; Rostami et al., 2019) followed the Generative Adversarial Networks (GANs) framework (Goodfellow, 2016) to produce generators for previous tasks, and learn parameters that fit a mixed set of real data of the new task and replayed/generated data of previous tasks. A network of experts is proposed by (Aljundi et al., 2016) to measure task relatedness to deal with catastrophic forgetting. Rannen Ep Triki et al. (2017) used an autoencoder to extend the LwF method (Li & Hoiem, 2016). Jung et al. (2016) proposed a less-forgetful learning that regularizes the final hidden activations. Rosenfeld & Tsotsos (2017) proposed controller modules that optimize loss on the new task with representations learned from previous tasks. They found that they could achieve satisfactory performance while only requiring about 22% of parameters of a fine-tuning method. Jin & Sendhoff (2006) modeled the catastrophic forgetting problem as a multi-objective learning problem and proposed a multi-objective pseudo-rehearsal framework to interleave base patterns with new patterns during optimization. Nguyen et al. (2017) proposed the variational continual learning by combining online variational inference (VI)

and Monte Carlo VI for neural networks. Seff et al. (2017) proposed to solve continual generative modeling by combining the ideas of GANs (Goodfellow, 2016) and EWC (Kirkpatrick et al., 2017).

Apart from regularization based approaches mentioned above (e.g., LwF (Li & Hoiem, 2016), EWC (Kirkpatrick et al., 2017) and their variants), dual-memory based learning systems have also been proposed. They are inspired by the complementary learning systems (CLS) theory (McClelland et al., 1995; Kumaran et al., 2016) in which memory consolidation and retrieval are related to the interplay of the mammalian hippocampus (short-term memory) and neocortex (long-term memory). Gepperth & Karaoguz (2016) proposed to use a modified self-organizing map (SOM) as the long-term memory. To complement it, a short-term memory (STM) is added to store novel examples. During the sleep phase, the whole content of STM is replayed to the system. This process is known as intrinsic replay or pseudo-rehearsal (Robins, 1995). It trains all the nodes in the network with new data (e.g., from STM) and the replayed examples/samples from previously seen classes or distributions on which the network has been trained. The replayed samples prevents the network from forgetting. A related work is that in Wu et al. (2018). Kemker & Kanan (2018) proposed a similar dual-memory system called FearNet. It uses a hippocampal network for short-term memory, a medial prefrontal cortex network for long-term memory, and a third neural network to determine which memory to use for prediction.

More recently, Hu et al. (2019) proposed a technique that learns to generate a part of the parameters while sharing the other part. Forgetting is dealt with by generating some previous parameters and regularization. Zeng et al. (2019) proposed a method that tries to learn the new task by revising the weights in the directions that are orthogonal to the data of the previous tasks so that the new task will not interfere with the old tasks. The work in Xu & Zhu (2018) searches for the best neural architecture for each coming task via reinforcement learning. Ritter et al. (2018) proposed an online Laplace approximation method. Dhar et al. (2019) proposed a method that combines three loss functions to encourage the model resulted from the new task (class) to be similar to the previous model. Some other related works include Learn++ (Polikar et al., 2001), Pathnet (Fernando et al., 2017), Memory Aware Synapses (Aljundi et al., 2017), One Big Net for Everything (Schmidhuber, 2018), Phantom Sampling (Venkatesan et al., 2017), Active Long Term Memory Networks (Furlanello et al., 2016), Conceptor-Aided Backprop (He & Jaeger, 2018), Gating Networks (Masse et al., 2018; Serrà et al., 2018), PackNet (Mallya & Lazebnik, 2017), Diffusion-based Neuromodulation (Velez & Clune, 2017), Incremental Moment Matching (Lee et al., 2017b), Dynamically Expandable Networks (Lee et al., 2017a; Yoon et al., 2018; Li et al., 2019), Incremental Regularized Least Squares (Camoriano et al., 2017), and Dual-Memory Recurrent Self-Organization (Parisi et al., 2018b). An excellent survey of continual learning can be found in (Parisi et al., 2018a).

Most continual learning works deal with CCL (class continual learning) where each task consists of one or more classes that are incrementally learned. This paper focuses on task continual learning (TCL), where each task is a separate classification problem. GEM (Gradient Episodic Memory) (Lopez-Paz & Ranzato, 2017) is such a method, which takes task id in addition to the training data of the specific task as input. GEM deals with forgetting when learning a new task by jointly training the current task using its training data and also a sampled data from the previous tasks to ensure the past knowledge is maintained. It also has a cross-validation step to tune the network using the first few tasks, which most other continual learning methods do not have. A-GEM (Chaudhry et al., 2019) significantly improved GEM's efficiency while achieving comparable or better results. Another recent work in this direction is 'learn to grow' (Li et al., 2019), which grows the network with each new task while encouraging sharing of network parameters using a new penalized loss function.

## 3 Model development

Traditionally, a neural network is trained to perform a single task. Training the network to perform an additional task causes catastrophic forgetting (CF) where the network is no longer able to perform the previous task well. CF is caused by the standard gradient descent method. We use learning of two tasks $k-1$ and $k$ to illustrate. Learning task $k$ in task $k-1$'s network can be formulated as

$$\boldsymbol{W}_l^{k,*} \leftarrow \boldsymbol{W}_l^{k-1,*} - \lambda \nabla \boldsymbol{W}_l^{k-1,*} \tag{1}$$

where $\lambda$ is the learning rate and $\boldsymbol{W}_l^{k,*}$ and $\boldsymbol{W}_l^{k-1,*}$ are the optimal parameters for task $k$ and task $k-1$ of layer $l = 0, \cdots, L$, respectively. From here, the subscript $l$ will be omitted since the analysis

is invariant with layers. When the model predicts for the previous task $k - 1$, it may not produce the optimal result for task $k - 1$ because

$$W^{k,*}x_{k-1} \neq W^{k-1,*}x_{k-1} \tag{2}$$

where $x_{k-1}$ is a sample from task $k - 1$ and $W^{k-1,*}$ is the optimal parameter for task $k - 1$.

### 3.1 PROJECTED GRADIENT UPDATE

Following the example above, if the parameters for learning task $k$ are updated in the directions perpendicular to the column space of data samples from the previous task, then forgetting can be avoided. This is formulated as

$$W^{k,*} \leftarrow W^{k-1} - \lambda \nabla W^{k-1} P^{k-1} \tag{3}$$

where $P^{k-1}$ is a projection matrix perpendicular to the column space of samples $X_{k-1}$. With this update, the new parameters for task $k$ also work for task $k - 1$ because

$$W^{k,*}x_{k-1} = W^{k-1}x_{k-1} - \lambda \nabla W^{k-1} P^{k-1} x_{k-1} = W^{k-1}x_{k-1} \tag{4}$$

Zeng et al. (2019) proposes the *orthogonal weight modification* (OWM) method which constructs $P$ by using the recursive least square algorithm. The recursive algorithm updates a linear model which gives the least square error of sequentially arriving data. Instead of discounting the previous errors, OWM discounts the current error and solves

$$\min \sum_{i=1}^{k-1} \|e(i)\|^2 - \alpha \|e(k)\| \tag{5}$$

where $e$ is the error and $0 < \alpha < 1$ is a parameter. By the matrix inversion lemma, it updates the projection for sample $x_k$ of task $k$ as (superscript $'$ means transpose)

$$P^k = P^{k-1} - (\alpha + x_k' P^{k-1} x_k)^{-1} P^{k-1} x_k x_k' P^{k-1} \tag{6}$$

and updates the weight as $W^{k+1} = W^k - \lambda P^k \nabla W^k$. Since $P^k$ projects $\nabla W^k$ onto the input space, the weight is modified within the subspace where a good performance was attained. By construction, if the optimal weight space for task 1 is span($W^1$), the updated $W^2$ for task 2 lies in span($W^1$) $\cap$ span($W^2$) $\neq \emptyset$. This makes sure that the model finds the solution without affecting the previous task performance, but this restricts the solution space for a new task. Thus, OWM is effective if tasks are similar. If the tasks are dissimilar, it will not learn well. For instance, in the extreme case, if span($W^3$) $\cap$ (span($W^1$) $\cap$ span($W^2$)) $= \emptyset$, the obtained $W^3$ will perform for task 1 and task 2, but will not perform for task 3. Even if there is intersection but the intersection is small, it will also be hard to find a good solution.

Therefore, an ideal projection should be such that 1) it is resistant to forgetting and 2) it does not restrict the solution space for the new task. Below, we propose a new projection method to achieve both objectives based on changing the data representation.

### 3.2 LEARNING WITH ORTHOGONAL REPRESENTATION

Let the data from task $k$ be $(X_k, t^k)$ where $X_k$ is the training data and $t^k$ is the indicator of task $k$. We use task number $k$ as the indicator, which is also used in our system. The proposed learning framework PCP is defined as follows.

For the data of task $k$, we change its representation with an orthogonal basis $\{c^{(i)}, \cdots, c^{(p)}\}$. We denote this basis matrix for task $k$ as $C^k$, and train the model using the transformed data $C^k X_k$. The projection matrix $P^k$ is constructed such that it maps to the space perpendicular to the current and all the previous orthogonal basis matrices $C^1, \cdots, C^k$. Let $C = [C^1, \cdots, C^k]$, the projection can be computed in a standard way as follows (again, superscript $'$ means transpose):

$$P^k = I - C(C'C)^{-1}C' \tag{7}$$

where $I$ is the identity matrix.

Suppose that the model is trained with the aforementioned framework from task 1 to task $k$. The parameter $\boldsymbol{W}^{k+1}$ for task $k+1$ is obtained by

$$\boldsymbol{W}^{k+1} = \boldsymbol{W}^k - \lambda \nabla \boldsymbol{W}^k \boldsymbol{P}^k \tag{8}$$

where $\nabla \boldsymbol{W}^k$ is the backpropagation with respect to $\boldsymbol{X}_{k+1}$. Then for a sample $\boldsymbol{x}_i$ from task $i \leq k$, we have

$$\boldsymbol{W}^{k+1} \boldsymbol{C}^i \boldsymbol{x}_i = \boldsymbol{W}^k \boldsymbol{C}^i \boldsymbol{x}_i - \lambda \nabla \boldsymbol{W}^k \boldsymbol{P}^k \boldsymbol{C}^i \boldsymbol{x}_i \tag{9}$$

$$= \boldsymbol{W}^i \boldsymbol{C}^i \boldsymbol{x}_i - \lambda \sum_{j=i}^{k} \nabla \boldsymbol{W}^j \boldsymbol{P}^j \boldsymbol{C}^i \boldsymbol{x}_i \tag{10}$$

$$= \boldsymbol{W}^i \boldsymbol{C}^i \boldsymbol{x}_i \tag{11}$$

$$= \boldsymbol{y}_i^* \tag{12}$$

where Eq. 10 is because $\boldsymbol{W}^k = \boldsymbol{W}^{k-1} - \lambda \nabla \boldsymbol{W}^{k-1} \boldsymbol{P}^{k-1} = \boldsymbol{W}^{k-2} - \lambda \sum_{j=k-2}^{k-1} \nabla \boldsymbol{P}^j = \cdots = \boldsymbol{W}^i - \lambda \sum_{j=i}^{k-1} \nabla \boldsymbol{W}^k \boldsymbol{P}^i$ and Eq. 11 is because $\boldsymbol{P}^j$ is a projection orthogonal to the basis matrices including $\boldsymbol{C}^i$.

In general, the data matrix $\boldsymbol{X} \in \mathbb{R}^{n \times d}$ spans the column space. This restricts the choice of $\boldsymbol{C}$ such that (1) it is an orthogonal matrix and (2) it sufficiently represents $\boldsymbol{X}$ with $m < d$ vectors, where $d$ is the dimensionality of the input vectors. We choose $\boldsymbol{C}$ to be $m$ principal component vectors of $\boldsymbol{X}$ as in *principal component analysis* (PCA). The new representation $\boldsymbol{Cx}$ captures the covariance, so it reduces the loss of information when the data is reduced to $m$ dimensions.

The projection matrix orthogonal to $\boldsymbol{C}$ ensures that the forward-pass returns optimal $\boldsymbol{y}^*$ if the input is represented in column space of $\boldsymbol{C}$. This is shown from Eq. 9 to Eq. 12. However, it does not restrict solution space because, by construction, the later projection is made of previous and new principal vectors so $\text{span}(\boldsymbol{P}^i) \subseteq \text{span}(\boldsymbol{P}^j)$ for $i < j$. In addition, during forward pass, $\boldsymbol{W}^{t+1} \boldsymbol{x} = \boldsymbol{W}^t \boldsymbol{x} - \nabla \boldsymbol{W}^t \boldsymbol{P}^t \boldsymbol{x}$, the projection maps input to the space orthogonal to columns of $\boldsymbol{C}^i$ for $i \in [1, t]$ and $\nabla \boldsymbol{W}^t$ also changes $\boldsymbol{P}^t \boldsymbol{x}$ to a different space. To put in another way, let the system learn a new task $k+1$, which follows a different distribution from task 1 to $k$ and cannot be expressed by the previous change of basis matrices $\boldsymbol{C}^1, \cdots, \boldsymbol{C}^k$. Then, there exists a basis $\boldsymbol{B}$ whose elements are linearly independent of $\boldsymbol{C}^1, \cdots, \boldsymbol{C}^k$. Denote $\boldsymbol{x}_{k+1}$ by a linear combination of vectors in $\boldsymbol{B}$. That is

$$\boldsymbol{x}_{k+1} = \boldsymbol{Ba} \tag{13}$$

for some coefficient vector $\boldsymbol{a}$. Since $\boldsymbol{P}^k$ is not orthogonal to $\boldsymbol{B}$, the forward-pass for sample $\boldsymbol{x}_{k+1}$ becomes

$$\boldsymbol{W}^{k+1} \boldsymbol{x}_{k+1} = \boldsymbol{W}^k \boldsymbol{x}_{k+1} - \nabla \boldsymbol{W}^k \boldsymbol{P}^k \boldsymbol{x}_{k+1} \tag{14}$$

$$= \boldsymbol{W}^k \boldsymbol{Ba} - \nabla \boldsymbol{W}^k \boldsymbol{P}^k \boldsymbol{Ba} \tag{15}$$

$$\neq \boldsymbol{y}_j^* \tag{16}$$

for any $j \in [1, k]$. Thus, the parameter will be updated during back-propagation to find $\boldsymbol{y}_{k+1}^* \neq \boldsymbol{y}_j^*$. That is, we are learning in a space beyond the existing solution space. We also note that if $\boldsymbol{x}_{k+1}$ is partially explained by previous principal vectors, i.e., $\boldsymbol{x}_{k+1} = \boldsymbol{C}^1 \boldsymbol{a}_1 + \cdots \boldsymbol{C}^k \boldsymbol{a}_k + \boldsymbol{Ba}_{k+1}$ for some unknown change of basis $\boldsymbol{B}$, then $\boldsymbol{P}^k \boldsymbol{x}_{k+1} = \boldsymbol{Ba}_{k+1} \neq 0$. Thus, the model learns the new part $\boldsymbol{Ba}_{k+1}$. These cases mean that the new task doesn't have to be learned in the existing solution space like OWM (Zeng et al., 2019), but separately. Thus the proposed method is able to learn better as we will see in the experiment section.

The detailed learning algorithm is given in Algorithm 1 and the testing process is given in Algorithm 2. In the algorithms, comments started with "//". With the comments, both algorithms are self-explanatory.[1]

## 4 EXPERIMENTS

In this section, we empirically evaluate the proposed method PCP and compare it with the latest state-of-the-art baselines.

---

[1]The principal component vectors are saved for each task during learning as they are also used in testing to transform the test data (see Algorithm 2).

---

**Algorithm 1** Principal Component Projection (PCP) training algorithm

---

**Require:** $\boldsymbol{P}_l^1 = \boldsymbol{I}$ for layers $l = 0, \cdots, L$ of task 1
 1: // The number of $\boldsymbol{T}$ tasks is unknown
 2: **for all** $k = 1, \cdots, \boldsymbol{T}$ **do**
 3:     // Train the model to correctly identify $\{(\boldsymbol{x}_k, k), y_k\}$
 4:     Compute loss with the standard forward pass
 5:     In backward pass, update parameters as $\boldsymbol{W}_l^k \leftarrow \boldsymbol{W}_l^k - \lambda \nabla \boldsymbol{W}_l^k \boldsymbol{P}_l^k$
 6:     // After forward pass, obtain the principal component vectors of neurons including input and hidden layers
 7:     // The number of principal vectors may vary for different layers (in our experiments, they are automatically selected based on the validate data)
 8:     // For example, choose $m_l$ principal vectors for the $n$ samples of activated neurons $\boldsymbol{A}_l \in \mathbb{R}^{n \times d}$ at $l$th layer (in our experiments, the number of principal vectors $m_l$ is also automatically selected based on the validation data)
 9:     **for all** $l = 0, \cdots, L$ **do**,
10:         Obtain principal component vectors $\boldsymbol{C}_l^k = \{\boldsymbol{c}^{(1),k}, \cdots \boldsymbol{c}^{(m_{l,k}),k}\}$
11:         // Denote the principal vectors from task 1 to $k$ at layer $l$ by $\boldsymbol{C}_l = [\boldsymbol{C}_l^1, \cdots, \boldsymbol{C}_l^k]$
12:         Construct $\boldsymbol{P}_l^k = \boldsymbol{I} - \boldsymbol{C}_l(\boldsymbol{C}_l' \boldsymbol{C}_l)^{-1}\boldsymbol{C}_l'$, where $'$ is transpose
13:         Change of basis $\boldsymbol{C}_l^k = [\boldsymbol{c}^{(1),k}, \cdots \boldsymbol{c}^{(m_{l,k}),k}, \boldsymbol{o}, \cdots, \boldsymbol{o}] \in \mathbb{R}^{d \times d}$, where $\boldsymbol{o}$ is the zero vector
14:     **end for**
15:     // Until convergence, train neural net $f$ to identify $\{(\boldsymbol{C}^k \boldsymbol{x}_k, k), y_k\}$
16:     In forward pass, apply $\boldsymbol{C}_l^k$ to input neurons of each layer.
17:     In backward pass, update parameters as $\boldsymbol{W}_l^k \leftarrow \boldsymbol{W}_l^k - \lambda \nabla \boldsymbol{W}_l^k \boldsymbol{P}_l^k$
18: **end for**
19: **return** $\boldsymbol{C}_0, \cdots, \boldsymbol{C}_L$

---

**Algorithm 2** PCP test algorithm

---

**Require:** The matrices, denoted by $\boldsymbol{C}_0, \cdots, \boldsymbol{C}_L$, of principal component vectors at each layer 0 to $L$
 1: // Retrieve corresponding principal vectors of task $k$ from $\boldsymbol{C}_l$ for each layer $l$
 2: // Make a prediction by changing the representations of input and activated neurons during forward pass
 3: In forward pass, apply $\boldsymbol{C}_l^k$ of $\boldsymbol{C}_l$ to input neurons of each layers.
 4: **if** $f(\boldsymbol{C}_0^k \boldsymbol{x}_k) == y_k$ **then**
 5:     Correct prediction
 6: **else**
 7:     Incorrect prediction
 8: **end if**

---

### 4.1 EXPERIMENT DATASETS

Four image classification benchmark datasets are used in our experiments.

(1). **MNIST** (LeCun et al., 1998).[2] This is a handwritten digit classification dataset of 10 digits. The dataset has 70,000 examples/instances. We use 60,000 for training and 10,000 for testing. 20% of the training data is also used as the validation set. Each class has the same number of examples in training, validation or testing as the dataset is balanced. This is also the case for the rest of the datasets.

(2). **CIFAR-10** (Krizhevsky & Hinton, 2009).[3] This is an image classification dataset consisting of 60,000 32x32 color images of 10 classes, with 6,000 images per class. We use 50,000 for training and 10,000 for testing. Again, 20% of the training data is used as the validation set.

---

[2]http://yann.lecun.com/exdb/mnist/
[3]https://www.cs.toronto.edu/ kriz/cifar.html

(3). **EMNIST-26** (Cohen et al., 2017).[4] EMNIST-26 consists of 26 letters, i.e., 26 balanced classes. It has 145,600 examples. We used 124,800 for training and the rest for testing. Again, 20% of the training data is also used as the validation set.

(3). **EMNIST-47** (Cohen et al., 2017).[5] EMNIST-47 consists of 47 characters, i.e., 47 balanced classes. It has 131,600 examples. We use 112,800 examples for training and the rest for testing. Again, 20% of the training data is used as the validation set.

## 4.2 BASELINE SYSTEMS

Our main baseline is the recent algorithm OWM (Zeng et al., 2019) published in Nature Machine Intelligence. We mainly compare with it due to two reasons. The first reason is that this is a strong baseline. Most of existing techniques for dealing with catastrophic forgetting (CF) tries to keep the weights learned for previous tasks unchanged or minimally changed so that the previously learned knowledge is not forgotten (see Section 2). This paper gives a novel and principled idea on how the previous weights can be kept while learning a new task. It also proposed a novel technique to implement the idea to deal with CF. The system OWM outperforms some recent techniques dramatically. For example, it outperforms the **PGMA** method in (Hu et al., 2019) by more than 10% (from 42.47% to 52.83%) for 5 tasks of CIFAR-10. PGMA already outperforms **IMM** (incremental moment matching) (Lee et al., 2017b) by 8%. The second reason is that our technique is inspired by their idea and it tries to address its shortcoming as discussed in Section 3.2. As we will see below, the idea in the paper is a good one. If the weakness is addressed, the results can be quite strong.

We also include the **EWC** method (Kirkpatrick et al., 2017) [6] as a baseline in comparison as EWC is commonly used in about all papers dealing with CF. We will not directly compare the proposed system PCP with PGMA and IMM as they are significantly inferior to OWM. All our results from OWM are obtained by running the code [7] released by the authors based on their tuned parameters. [8]

Although OWM works in the CCL (class continual learning) setting (not task continual learning (TCL)), it is easy to use it in the TCL setting by restricting testing of each test instance to the classes of the specific task. That is, if a test instance is from task 1, we will only use the final classification nodes corresponding to the classes of task 1 to predict the class of the test instance. Classifications of other nodes are ignored regardless what their values may be for the test instance. Thus, in the TCL setting the classification results are usually better than those of the CCL setting because the task id is known for each test instance in the TCL setting. For a fair comparison, the proposed PCP method uses exactly the same network and the same network size to learn and to test. As mentioned above, we also run their released code to produce the results reported below.

## 4.3 TRAINING DETAILS

Since we mainly compare with OWM, we use the same architecture as OWM. For MNIST, we use 3 fully connected layers with size $784 - 800 - 800 - 10$. For CIFAR-10, we used Alexnet as a pre-trained feature extractor, which is fixed through out the training. Fully connected layers of sizes $9216 - 4096 - 1000 - 1000 - 10$ are used for classification. 9216 is the number of output neurons of the feature extractor. OWM uses the same pre-trained feature extractor. For EMNIST-26 and EMNIST-47, we used fully connected layers of sizes $784 - 800 - 800 - 26$ and $784 - 800 - 800 - 47$, respectively. We did not use convolution layers for MNIST and EMNIST because the two datasets consist of single channel images and the model could attain high accuracy without using it. We used stochastic gradient descent with minibatch size of 40, 64, and 40 for MNIST, CIFAR-10, and EMNIST, respectively. A momentum value of 0.9 was chosen for better convergence. Regularization methods such as dropout were not used in any model.

---

[4]https://www.nist.gov/node/1298471/emnist-dataset

[5]https://www.nist.gov/node/1298471/emnist-dataset

[6]https://github.com/moskomule/ewc.pytorch

[7]https://github.com/beijixiong3510/OWM

[8]We are unable to compare with the recent method in (Li et al., 2019) as its code was not released and we also did not manage to get the code from the authors via email. Systems that store sampled original data of previous tasks (Chaudhry et al., 2019; Rebuffi et al., 2017) are not comparable with us. These systems also require a pre-cross-validation phase using the first few tasks to pre-set the network. We don't need either.

Table 1: Average accuracy over all tasks after the last task is learned

| Model | MNIST (5 tasks) | CIFAR10 (5 tasks) | EMNIST-26 (5 tasks) | EMNIST-26 (10 tasks) | EMNIST-47 (5 tasks) | EMNIST-47 (10 tasks) |
|---|---|---|---|---|---|---|
| EWC | 96.14 | 67.33 | 51.16 | 42.83 | 48.98 | 52.11 |
| OWM | **98.89** | 78.29 | 42.04 | 50.00 | 24.61 | 38.39 |
| PCP | 98.54 | **83.43** | **71.88** | **74.25** | **65.61** | **74.60** |

For the experiments with PCP, we used $20\%$ of training data as the validation set to automatically select a good set of parameters, which includes the number of epochs, learning rates and the number of principal vectors at each level.

## 4.4 TEST RESULTS AND ANALYSIS

The experimental results in accuracy are reported in Table 1, which are obtained as follows. We first incrementally learn each task one by one. After all tasks are learned, we test all tasks using their respective test data. As noted above, the parameters and training epochs are selected using the validation set of each task. The test results are given in Table 1.

In Table 1, the first row gives the dataset names and number of tasks for each dataset. For MNIST and CIFAR-10 of 5 tasks, each task consists of 2 classes. For example, for MNIST, the tasks are sets of two digits/classes, i.e., {0, 1}, {2, 3}, {4, 5}, {6, 7}, and {8, 9}. Since EMNIST-26 has 26 classes, the 5 tasks consist of 5, 5, 5, 5, and 6 classes. For 10 tasks, they have 2, 2, 2, 2, 3, 3, 3, 3, 3, and 3 classes. For EMNIST-47, the 47 classes are split into 10, 10, 10, 9, and 8 classes for 5 tasks, and 3, 4, 5, 5, 5, 5, 5, 5, 5, and 5 classes for 10 tasks.

From Table 1, we can observe that PCP consistently outperforms baselines on all datasets and tasks except MNIST, for which OWN and PCP are about the same. All three systems perform well on MNIST as it is a relatively easy dataset. Although OWM performs well for CIFAR-10, it does poorly for EMNIST-26 and EMNIST-47. We tuned OWM extensively, but were not able to obtain better results. We believe that it is due to the restricted solution space weakness of the system identified in this paper (Section 3.2). In the case of CIFAR-10, the solution spaces of the tasks are similar for OWM as ImageNet contains a large number of similar images as those in CIFAR-10. However, ImageNet feature extractor is not so useful for EMNIST as ImageNet does not have similar images to handwritten characters from EMNIST (thus not used), which causes OWM to fail due to the weakness identified. Note that CIFAR-10's result reported here for 5 tasks is better than that in the OWM paper because the TCL setting is easier than the CCL setting, as at the test time, the system knows the task id of the test instance, but in the CCL setting, that knowledge is not available.

Note that for both EMNIST-26 and EMNIST-47, the results for 5 tasks are poorer than for 10 tasks. This is because for the 10 tasks case, each task has fewer classes and it is easy to get higher accuracy with fewer classes.

**Execution time**: The proposed system PCP is about 1.5 times slower than OWM. The main reason is that PCP needs to train the mapping from $x$ to $y$ and from $Cx$ to $y$. The projection matrix requires a matrix inversion. This needs to be computed once after a task is learned. Obtaining the principal vectors requires computing eigenvectors. It is also computed only once after training. EWC is about 5 times slower than OWM.

## 5 CONCLUSION

To make a deep neural network model learn new tasks continually or incrementally, we have to deal with catastrophic forgetting. This paper proposed a novel method, called *principal components projection* (PCP), which is based on the idea that if network weights are updated only for the part of training data that the model has not seen in the orthogonal directions to those of previous tasks, interfering with the previously learned knowledge can be largely avoided. The proposed method works by transforming the input vectors and activated neurons for each task using orthogonal basis matrices, which are lists of principal components of the corresponding neurons. We modify the gradient if the sample does not lie on the column space spanned by all the previous task data. Experimental results showed superior performance of the proposed method PCP compared to the

existing state-of-the-arts. Our future work will focus on further improving the accuracy and studying how to enable and exploit knowledge sharing across tasks to improve the performance of all tasks.

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
