# OpenReview forum: "Continual Learning via Principal Components Projection"
_ICLR.cc/2020/Conference — Reject_

### Official Review · AnonReviewer1 · 2019-10-22
**Official Blind Review #1**

**Rating:** 3

**Review:**

The method proposes a method for continual learning. The method is an extension of recent work, called orthogonal weights modification (OWM) [Zheng,2019]. This method aims to find gradient updates which are perpendicular to the input vectors of previous tasks (resulting in less forgetting). However, the authors argue, that the learning of new tasks is happening in the solution space of the previous tasks, which might severely limit the ability to adapt to new tasks. The authors propose a ‘principal component’-based solution to this problem. The method is considering the ‘task continual learning’ scenario (also known as task-aware) which means that the task label is given at inference time.

Conclusion:

1. The paper is not well-positioned in related works. I think the work is more related to works with ‘parameter isolation methods’ such as Piggyback, Packnet, HAT. These methods reserve part of the capacity of the network for tasks. I think the authors should relate their work with these methods, and provide an argument of the problem with these previous methods, which is addressed by their approach. I can see that rather than freezing weights (PackNet) or features (HAT) , the method freezes linear combinations of features. But it is for me not directly clear that that is desirable. In HAT the backpropagated vector is projected on the mask vector which coincides with the neurons (activations).

2. The experimental verification of the paper is too weak, and only comparison to EWC and OWM (not well known) are provided. At least a comparison with the more related works PackNet and HAT should be included. For more recent method for task-aware CL see also ‘Continual learning: A comparative study on how to defy forgetting in classification tasks’. Also results seem bad. For example on CIFAR10, 5 tasks in TCL setting is two-class problem per task; I would expect better results.

3. The authors claim that OWM is effective if tasks are similar, but not when dissimilar. And the proposed PCP solves this problem. However, all experiments are on similar tasks, and no cross domain tasks are considered, e.g. going from MNIST (task1) to EMNIST-26 (task2) etc. This would empirically support the claim. Also, the authors expect the difference between PCP and OWM to be even larger then.

4. Some more analysis of the success of PCA in representing the distribution would be appreciated, e.g. the percentage of total energy which is captured (sum of selected eigenvalues divided by sum of all eigenvalues). Such an analysis of P_l^k as a function of the tasks (and for several layers) would be interesting to see, for example for EMNIST-47(10 tasks).

5. Novelty with respect to OWM is rather small.

6. The authors should mention that the method is pretrained on ImageNet in section 4.3. Given these datasets, I think it makes more sense to train from scratch and I would like to see those results.

Minor remarks:
- I wonder if you use OWM or PCP you discard the possibility of positive backward transfer. Maybe the authors could comment on that.

- The authors write that ‘TCL setting the classification results are usually better than those of the CCL’ is that not per definition true ? Anything correctly classified under CCL is correctly classified under TCL but not the other way around.


**Experience Assessment:**

I have published one or two papers in this area.

**Review Assessment: Checking Correctness Of Derivations And Theory:**

I assessed the sensibility of the derivations and theory.

**Review Assessment: Checking Correctness Of Experiments:**

I assessed the sensibility of the experiments.

**Review Assessment: Thoroughness In Paper Reading:**

I read the paper at least twice and used my best judgement in assessing the paper.

---

### Official Review · AnonReviewer2 · 2019-10-22
**Official Blind Review #2**

**Rating:** 3

**Review:**

This paper introduces Principal Components Projection, a method that computes the principal components of input vectors, using them to train on a transformed input space and to project gradient updates. Experiments show improved results over OWM (the method that this paper builds on) and EWC.

If I understand correctly (which I think may not be the case), the principal component vectors are computed after the first forward/backward pass of each task, for the inputs to each layer (C_l^k). These principal components are then fixed, the orthogonal projection matrix P_l^k is then found, and then normal training is iterated until convergence using this C_l^k and P_l^k.

Questions:
- Seeing as (especially for the first task), weights are initialised randomly, why does this method provide reasonable principal components for layers after the first layer?
- I also do not understand why the dxd projection matrix P, which is orthogonal to all previous basis matrices C, has the property span(P^i) \subset span(P^j) for i < j. Surely as more basis matrices are found, then the orthogonal space restricts in size.
- I also do not understand Equation 1. What is \grad{W}? If it is, as defined 2 pages later, 'the backpropagation with respect to X_{k+1}' [or X_k here], then is Equation 1 saying that only one gradient step is used per task?

The experiments seem reasonable, except that there are no standard deviations on the results. However, as far as I'm aware, these experimental protocols (dataset and model size) are not used in other papers: it would be nice to see experiments which match previous papers' protocols, for example with MNIST and CIFAR-10 at least (other papers use smaller model sizes).

As it is currently, I am unable to understand the paper despite spending some time trying to understand it. I am therefore giving the paper a weak reject. Hopefully the authors can answer my questions.

Finally, some minor specific suggestions for improving the writing:
- Immediately after Equation 12, there is \grad{P^j} instead of \grad{W^j}{P^{k-2}}
- The paragraph before Equation 13 uses 't' instead of 'k' sometimes for task index
- Use `   not ' for open quotation marks

**Experience Assessment:**

I have published one or two papers in this area.

**Review Assessment: Checking Correctness Of Derivations And Theory:**

I assessed the sensibility of the derivations and theory.

**Review Assessment: Checking Correctness Of Experiments:**

I assessed the sensibility of the experiments.

**Review Assessment: Thoroughness In Paper Reading:**

I read the paper thoroughly.

---

### Official Review · AnonReviewer3 · 2019-10-24
**Official Blind Review #3**

**Rating:** 3

**Review:**

This paper tries to solve the catastrophic forgetting issue in the continual learning problem. The authors propose a method based on principal components projection to tackle this issue.  The authors conduct experiments on image classification tasks to show the performance of the proposed method and compare it with two other baselines EWC and OWM.

Strong points:
1. This paper tries to solve an important problem.
2. The intuition of applying principal components projection is straightforward.

Weak points:
1. The most concerned point about this paper is the experiment. It is not convincing. The authors claim that OWM is one of the strongest baselines, but actually it perform really badly on EMNIST-26 (5 tasks),  EMNIST-47 (5 tasks) and EMNIST-47 (10 tasks). What is the reason? Is it because of insufficient parameter tuning? If different methods perform differently on various datasets, it is really necessary to show more baseline methods to illustrate that the proposed method has universally good performance on different datasets.
2. It might strengthen the paper if the authors can show the comparison results on more other datasets, e.g., other image classification tasks. It would be better if the authors can show the proposed method can generalize to other tasks.
3. The authors point out that one key drawback of OWM is that, if the tasks are not quite related, OWM may perform very badly. Is this the case in the experiment?
4. It is not clear why the proposed method can solve the issue that OWM faces with (bad accuracy when tasks are not quite related).


**Experience Assessment:**

I have read many papers in this area.

**Review Assessment: Checking Correctness Of Derivations And Theory:**

I assessed the sensibility of the derivations and theory.

**Review Assessment: Checking Correctness Of Experiments:**

I carefully checked the experiments.

**Review Assessment: Thoroughness In Paper Reading:**

I read the paper at least twice and used my best judgement in assessing the paper.

---

### Decision · Program_Chairs · 2019-12-19

**Decision:**

Reject

**Comment:**

There is no author response for this paper. The paper addresses the issue of catastrophic forgetting in continual learning. The authors build upon the idea from [Zheng,2019], namely finding gradient updates in the space perpendicular to the input vectors of the previous tasks resulting in less forgetting, and propose an improvement, namely to use principal component analysis to enable learning new tasks without restricting their solution space as in [Zheng,2019].
While the reviewers acknowledge the importance to study continual learning, they raised several concerns that were viewed by the AC as critical issues: (1) convincing experimental evaluation -- an analysis that clearly shows how and when the proposed method can solve the issue that [Zheng,2019] faces with (task similarity/dissimilarity scenario) would substantially strengthen the evaluation and would allow to assess the scope and contributions of this work; also see R3’s detailed concerns and questions on empirical evaluation, R2’s suggestion to follow the standard protocols, and R1’s suggestion to use PackNet and HAT as baselines for comparison;  (2) lack of presentation clarity -- see R2’s concerns how to improve, and R1’s suggestions on how to better position the paper.
A general consensus among reviewers and AC suggests, in its current state the manuscript is not ready for a publication. It needs clarifications, more empirical studies and polish to achieve the desired goal.